# A Machine Learning Approach to Screen for Otitis Media Using Digital Otoscope Images Labelled by an Expert Panel

**DOI:** 10.3390/diagnostics12061318

**Published:** 2022-05-25

**Authors:** Josefin Sandström, Hermanus Myburgh, Claude Laurent, De Wet Swanepoel, Thorbjörn Lundberg

**Affiliations:** 1Department of Public Health and Clinical Medicine, Unit of Family Medicine, Umeå University, 901 87 Umeå, Sweden; josefin.sandstrom@umu.se; 2Department of Electrical, Electronic and Computer Engineering, University of Pretoria, Pretoria 0110, South Africa; herman.myburgh@up.ac.za; 3Department of Public Health and Clinical Medicine, Unit of Medicine, Umeå University, 901 87 Umeå, Sweden; claude.laurent@umu.se; 4Department of Speech-Language Pathology and Audiology, University of Pretoria, Pretoria 0110, South Africa; dewet.swanepoel@up.ac.za

**Keywords:** artificial intelligence, machine learning, convolutional neural network, otitis media, global health, digital imaging

## Abstract

Background: Otitis media includes several common inflammatory conditions of the middle ear that can have severe complications if left untreated. Correctly identifying otitis media can be difficult and a screening system supported by machine learning would be valuable for this prevalent disease. This study investigated the performance of a convolutional neural network in screening for otitis media using digital otoscopic images labelled by an expert panel. Methods: Five experienced otologists diagnosed 347 tympanic membrane images captured with a digital otoscope. Images with a majority expert diagnosis (*n* = 273) were categorized into three screening groups Normal, Pathological and Wax, and the same images were used for training and testing of the convolutional neural network. Expert panel diagnoses were compared to the convolutional neural network classification. Different approaches to the convolutional neural network were tested to identify the best performing model. Results: Overall accuracy of the convolutional neural network was above 0.9 in all except one approach. Sensitivity to finding ears with wax or pathology was above 93% in all cases and specificity was 100%. Adding more images to train the convolutional neural network had no positive impact on the results. Modifications such as normalization of datasets and image augmentation enhanced the performance in some instances. Conclusions: A machine learning approach could be used on digital otoscopic images to accurately screen for otitis media.

## 1. Introduction

Otitis media includes several commonly occurring inflammatory conditions of the middle ear with potentially severe effects if untreated. The disease spectrum ranges from benign inflammatory conditions to more severe and even destructive chronic middle ear disease with life threatening complications [1,2]. The key to effective treatment of otitis media, whether simple or severe, is to establish a correct diagnosis. However, doctors who can diagnose these conditions correctly are scarce in a substantial part of the world. For example, there are only slightly more than one Ear-, Nose-, and Throat (ENT) specialist per million people in sub-Saharan Africa [3]. As a measure, there is an increasing emphasis in the field of global health to task-shift by assigning some of the work traditionally performed by doctors to other categories of health care workers [4,5]. Digital imaging and telemedicine can be used by health care workers and diagnosis can then be made remotely by doctors [6]. This technique has opened up the possibility of adding assistive diagnostic tools based on digital imaging.

Early investigations in this field demonstrated that otitis media could be automatically diagnosed using image processing and decision tree classifications [7,8]. These systems were fixed and could not be modified. Later systems instead utilised machine learning, a branch of artificial intelligence (AI) with the capability to learn and improve from experience. The use of AI is currently studied from various perspectives within the medical field, e.g., to predict progression of disease and cancer survival [9,10]. This technology is also evolving rapidly in otology, and various types of AI systems have been developed and tested for the purpose of diagnosing ear disease [11,12,13,14,15,16,17]. These studies looked at different aspects of various ear conditions and the overall accuracy varied between 84% and 94% [11,12,13,14,15,16,17]. Convolutional neural network (CNN) is one kind of machine learning which utilises deep learning to discover structures in a dataset. It is modelled on the function of the mammalian visual cortex and can typically be used for image interpretation [18].

It is well-known that diagnosing otitis media correctly is difficult. Professionals involved as part of their daily clinical work identify, as an example, otitis media with effusion (OME) correctly in 43–52% of cases [19,20] and acute otitis media (AOM) correctly in 62–78% of cases [21,22]. An AI system that could perform similarly or even better than the first line of doctors (often general practitioners or pediatricians) could be invaluable for triage of patients for appropriate levels of medical attention, cross-checking of results, and in severely under-sourced settings providing the only means of diagnosis.

The aim of this study was to investigate the performance of a machine learning approach (CNN) for otitis media screening using digital otoscopic images of tympanic membranes diagnosed by an expert panel.

## 2. Materials and Methods

In this prospective study we collected digital images of tympanic membranes and used an expert panel’s diagnoses of the images as reference standard in a comparison to the diagnoses of the same images by a CNN. Ethical clearance was provided by the Ethics committee at the University of Pretoria, South Africa, and for data analyses in Sweden by the Ethics committee at Umeå University. Data collection took place between March and May 2016. Written informed consent was obtained from the participants or their parents or legal guardians.

### 2.1. Participants

Enrolment of participants took place in low-income communities in the City of Tshwane (Pretoria), Gauteng, South Africa at one district hospital ENT clinic, three primary healthcare clinics and one itinerant clinic for school screening. Anyone in the waiting room could participate in the study, irrespective of ear complaints or not. Exclusion criteria were active draining of ears and persons younger than one year of age. The reason why actively draining ears were excluded was that some participants were also part of an audiological study (reported elsewhere) for which dry ears was an inclusion criterion. All participants gave their written informed consent to participate. All procedures complied with the Declaration of Helsinki.

### 2.2. Collecting Tympanic Membrane Images

Images were captured with a handheld digital otoscope (Welch Allyn Digital Macroview Otoscope, Welch Allyn Inc., Skaneateles Falls, New York, NY, USA). A series of images was collected from each ear, the number depending on how well all parts of the tympanic membrane could be visualised. The otoscope was connected to a PC laptop running Microsoft Windows 8 on which images were saved. Different sizes of ear specula were available to fit different widths of ear canals. A medical intern (JS) with prior training and experience in otoscopy performed all the data collection. A total of 380 ears were included in the study.

### 2.3. Expert Panel

In order to ascribe each ear a diagnosis (reference standard) an expert panel was appointed. Five otologists (two from South Africa and three from Sweden) with otological experience varying between 20 and 40 years agreed to participate. An odd number of panel members was chosen to enable majority decisions.

The diagnostic categories in this study were the following: normal tympanic membrane (Normal), acute otitis media (AOM), otitis media with effusion (OME), chronic suppurative otitis media (CSOM), obstructing wax (Wax) and not possible to determine (NPD). The CSOM group in this study refers to dry chronic perforation (inactive CSOM), since wet ears were excluded.

Before the expert panel assessment, a review of the series of images from the 380 ears was performed (Figure 1). Two of the authors, TL and CL, both with >20 years’ experience in otologic diagnostics, selected two to five of the best images from each series and transferred them to a graphic interchange format file (GIF file). Good image quality and visualisation of most of the tympanic membrane were the criteria for selection. In addition, the single best image was chosen from each ear for later classification by the CNN. Thirty-three ears did not reach the expert panel due to reasons given in Figure 1. A Google form was developed for the expert panel’s assessment, in which the GIF files were uploaded as separate cases together with three questions to be answered for every case: 1. Diagnosis (six alternatives to choose from, see above). 2. Confidence—How confident are you on your chosen diagnosis (unsure, confident, or very confident)? 3. Image quality—How do you assess the image quality of the presented tympanic membrane images (poor, acceptable or excellent)?

Briefing with the expert panel members was carried out via dispatched definitions of diagnoses (Appendix A), based on established diagnostic criteria [23,24] together with information about how to conduct the assessments. A calibration session was then launched, with a pilot sample of 20 GIF files, presented in an online Google form. The answers from the calibration were reviewed by one of the researchers (TL). The panellists’ assessments were analysed regarding any discordance in their assessments and/or misunderstandings. An instructive email on these findings with clarifications was then returned. A second calibration session was thereafter performed using seven images with previous disagreements and, after satisfactory alignment to the definitions, the expert panel was considered to be fully calibrated.

The assessment of the 347 ears in the main image set (Figure 1) was performed in identical Google forms as for the calibration. Independently of each other, the panel members made their assessment of each GIF-file. To avoid loss of concentration they assessed a randomised selection of <50 GIF files at a time, in eight subsequent sessions. Not until all five panellists had completed one session was the next made available. A reference standard diagnosis was established when three or more of the five panellists did agree on a diagnosis. The expert panel’s assessment rendered a reference standard of 311 ears, since for 36 ears a majority decision could not be reached (Figure 1). The 311 ears were diagnosed as 183 Normal, 2 AOM, 22 OME, 20 CSOM, 46 Wax and 38 NPD.

To prepare for the CNN analyses, a new grouping of diagnoses was made into three basic screening categories: Normal, Pathological (AOM, OME and CSOM) and Wax. Cases with the diagnosis NPD could not be classified into any of these categories and were therefore transferred to a separate group for later sub-group analysis (Figure 1). The previously chosen single best image from each of the remaining 273 GIF files constitute dataset 1 (see below).

A secondary target of this study was to investigate a potential effect of larger datasets for the performance of the CNN, and thus a second dataset was introduced which contained images from various imaging devices (described below).

### 2.4. Data Description & Preparation

Two different datasets of tympanic membrane images were used in this study.

Dataset 1: The images in dataset 1 derive from the process described above. The process resulted in 273 images, each with a resolution of 1280 × 1024 pixels and containing the following number in each diagnostic group: 183 Normal, 44 Pathological and 46 Wax.

Dataset 2: This dataset is identical to the one used in Myburgh et al. [11]. It consists of 389 tympanic membrane images acquired from numerous video-otoscopes and rigid endoscopes with diagnostic agreement between two independent expert examiners. The image resolution varies within the dataset. The dataset included 123 Normal, 206 Pathological and 60 Wax.

Pre-processing: All images in both dataset 1 and dataset 2 were cropped to ensure that black areas on the sides were removed. No colour correction or colour normalization was made.

Normalization: A trained CNN is biased towards classes with higher image numbers, resulting in lower diagnostic accuracy for classes with lower image numbers. To mitigate this effect, image numbers were normalized by randomly removing images from classes with higher image numbers. The number of images in each diagnostic group in the normalized data sets is equal to the group with the lowest number of images. Dataset 1 was normalized to a new dataset, dataset 1_norm_, so that each class contained 44 images, while dataset 2 was normalized to the dataset 2_norm_, so that each class contained 60 images. Analysis was performed on both the non-normalized and the normalized datasets.

Augmentation: Since a CNN requires large datasets for effective training, the number of images in dataset 1 and dataset 2, as well as dataset 1_norm_ and dataset 2_norm_, were increased by using a technique called image augmentation. Images of high resolution can be used to extract a number of low-resolution images. For example, an image of 320 × 320 high-resolution can be divided in 10 rows and 10 columns and generate ten 32 × 32 low-resolution images. From the 1024 (32 × 32) separate blocks, 1024 pixels are randomly sampled, one per block. Each sampled pixel is assigned to the new image in the pixel position corresponding to the block it was sampled from. The resulting images will look like the original image as well as to one another, but they will be unique images as they do not c contain shared pixels. Figure 2 demonstrates this concept. Analysis was performed with different augmentation factors (0, 5, 10, and 20). For instance, an augmentation factor of 5 means that five low resolution images were created from each high-resolution image as explained above.

### 2.5. CNN Description

The CNN that was used is known as Inception v1, also known as GoogLeNet, reported in Szergedy et al. [25]. This CNN model is available to freely use as part of Google’s Tensorflow platform. It consists of 22 layers made up of various convolution, pooling, and dense layers. It also makes use of a so-called “inception module” to assist with dimensionality reduction in the shallow layers in order to ease computation in the deeper layers.

Figure 3 shows the structure of this CNN. It consists of structure combinations of convolutional layers (large blue rectangles), dense layers (green), pooling layers (red), and activation/decision layers (yellow). Figure 4 shows the inception module architecture with its various convolutional and pooling blocks.

The standard Tensorflow implementation was used without any architectural modification. The batch size was 128. It used asynchronous stochastic gradient descent for training with an initial learning rate of 0.1 and an exponential decay factor of 0.1. The learning rate was updated every 50 epochs. Image data were reformatted to conform to the required input data format.

### 2.6. Scenarios for Training and Testing

Training was performed using a random selection of 75% of the images in each dataset, and testing was carried out on the remaining 25%. The CNN was trained and tested using different combinations of training and testing data in four different scenarios, A–D. For scenario A and B, only dataset 1 was used, while scenarios C and D investigated the effect of a larger dataset. Each scenario is described below.

A: Training and testing on dataset 1.

B: Training and testing on dataset 1_norm._

C: Training on dataset 1 and dataset 2 combined and testing on dataset 1.

D: Training on dataset 1_norm_ and dataset 2_norm_ combined and testing on dataset 1 _norm._

Instead of normalizing the datasets (or under-sampling by removing images) and applying data augmentation (a form of oversampling by creating new images) as described in Section 2.4, an alternative approach could have been taken to ensure that no data are discarded. Instead of data resampling, the classifier could have been modified by applying thresholding, learning rate adjustment or adjustment of prior class probabilities. It has been shown that oversampling outperforms the other class modification methods and in cases of large class imbalance ratio under-sampling is on par with oversampling [26]. Since both data under-sampling and oversampling were applied, accuracy is expected to be comparable to or slightly better than classifier modification.

#### Statistical Analysis

Otologists’ accuracy approximates to 95% in diagnosing OME [27]. For this study, we set the accuracy of the automated classification software as at least 85%, which resulted in an estimated sample size of 142 ears.

To assess the performance of the CNN, a confusion matrix was calculated for each scenario for the different augmentation factors. A confusion matrix is a means of visualising several important quantitative performance measures—true positive rate (TPR), false positive rate (FPR), true negative rate and false negative rate—for all diagnostic groups, by comparing actual diagnoses (reference standard) with the CNN diagnoses. The overall accuracy for each situation was also calculated.

To evaluate the CNN’s screening performance, sensitivity and specificity were calculated using an online calculator, MedCalc (© 2020 MedCalc Software Ltd. https://www.medcalc.org/calc/diagnostic_test.php (accessed on 22 September 2021). To enable these calculations, a division of the tympanic membrane images was made into normal and abnormal (Pathological and Wax).

Calculations were also performed on ratings from the expert panel members regarding their confidence in each diagnosed image and their apprehension of the image quality.

A separate sub-analysis was performed to investigate how the CNN classified the NPD images.

## 3. Results

For images presented to the CNN (*n* = 273), full agreement on the diagnosis (five out of five experts) was reached in 49% of cases (Table 1). In 24%, four out of five agreed and in the remaining 27% three out of five agreed. How the agreement differed across diagnoses is shown in Table 1. In these 273 images the expert panel members were confident or very confident in their diagnoses in 80% of the cases (and unsure in 20%). The image quality was rated as acceptable or excellent for 67% of the images.

### 3.1. Scenarios A and B

Overall accuracy for scenario A and B was generally high, and above 0.9 in all cases except one (scenario B without augmentation) (Figure 5). The highest overall accuracy (1.0) was achieved in scenario A without augmentation. Neither normalization nor augmentation improved the overall accuracy for dataset 1. In scenario B, augmentation at a moderate degree (factor 5 and 10, but not 20) did enhance the overall accuracy.

For all augmentation factors in scenario A, the TPRs were above 0.78 (Table 2). For Normal, this was 1.0 for all augmentation factors. The only notable misclassification rates (≥0.10) in scenario A were for Pathological, misclassified as Normal for augmentation factors 10 and 20 and as Wax for augmentation factor 5. As seen in Table 3, the sensitivity in these instances had dropped from 100% to 95% and 93% respectively, but specificity was still 100%.

For scenario B, the TPRs were above 0·72 for all augmentation factors (Table 2). The FPRs were above 0·10 in five instances, Pathological being misclassified as Wax (augmentation factors 0, 5 and 20) and Wax being misclassified as Pathological (augmentation factor 10 and 20).

When evaluating the effect of normalization in terms of TPR improvement (comparing scenarios A and B), normalization did in three instances improve the TPR, in five instances there were no difference, and in four instances the TPR was lowered in the normalized dataset.

While augmentation degraded the TPR for Pathological in scenario A (Table 2), it enhanced TPR in scenario B with the peak at augmentation factor 10 (0.96) (Table 2) and. also enhanced the overall accuracy for scenario B from 0·88 to 0·93 (Figure 5).

### 3.2. Scenarios C and D

Adding dataset 2 to the training set rendered an overall accuracy ranging between 0·92 (scenario C, augmentation factor 5), and 0·98 (scenario C, augmentation factor 20) (Figure 5). In scenarios C and D, the TPRs were still 1.0 for all augmentation factors for Normal but ranged from 0·59 to 0·92 for Pathological (Table 4).

### 3.3. Sub-Analysis

A sub-analysis was performed on the nine images for which all expert panel members had agreed upon the NPD classification (Table 1). Reasons reported for choosing the NPD category included poor image quality and low illumination. These nine images together with the respective diagnoses made by the CNN using the highest overall accuracy (dataset 1, no normalization, no augmentation) are seen in Figure 6. No NPD ears were classified as Normal by the CNN.

## 4. Discussion

This study demonstrates accurate screening for otitis media using a machine learning approach on digital otoscopic images diagnosed by an expert panel. A CNN was used, and the highest overall accuracy was found for dataset 1 without normalization and augmentation. When adding more images for training of the CNN (scenarios C and D), the overall accuracy slightly decreased, but sensitivity and specificity was still high, also with normalized datasets and across augmentation factors.

To our knowledge, within the field of machine learning in otology this study is unique in terms of its robust reference standard. In a recent meta-analysis on artificial intelligence in diagnosing ear disease by Habib et al., the authors concluded that ground truth assessment is a widespread limitation of existing studies, with limited studies using multiple independent experts’ diagnoses [28]. We used five experienced otologists to assess the images and required a minimum of three of them to agree upon the diagnosis to incorporate an image into the reference standard. In comparison, Livingstone et al. used two ENT specialists’ consensus diagnoses [12], and in a study by Cha et al. the main author labelled all images, using information from the medical record to double check [14]. A CNN is trained on labelled images and a less solid training material will affect the accuracy of the output.

The expert panel members did not always agree about the diagnosis, which reflects the difficulty in diagnosing tympanic membranes using otoscopy, nor were they confident in their own diagnosis in every case. In 10% (36/347) of cases a majority decision could not be reached. A study by Kuruvilla et al. showed poor agreement between three experts when assigning reference standard diagnoses [7]. In a re-assessment, their panel members were also asked to register their confidence in the chosen diagnosis and only images in which they all agreed and could report a certain confidence level were used in their ground truth set. This process reduced the number of images in their study considerably, from 826 to 181. For our reference standard, we did no discrimination based on confidence in the diagnosis. On the other hand, we employed more panel members and gave the alternative of NPD, most likely eliminating some cases with uncertain diagnoses, and thus maintaining a high quality reference standard.

The overall accuracy dropped when augmentation and normalization were used on dataset 1. Normalization rendered a reduction of images in the larger classification groups, in our dataset ‘the Normal’. The change in the proportions of diagnoses may have influenced the performance of the CNN in scenario B. Training and testing images were randomly sampled from the dataset and the difference between scenario A and B could theoretically have been a result of coincidence in the sampling. However, since the pattern was evident for all augmentation factors it seems more probable that the change in proportions of diagnoses was the reason. Diagnostic agreement between expert panel members was higher for Normal than for OME. It may be that Normal are easier to identify for the CNN and reducing the number of Normal through normalization decreases the accuracy of the CNN.

Our results show that augmentation of dataset 1_norm_ enhanced the overall accuracy of the CNN up to an augmentation factor of 10, but with higher augmentation it decreased again. The TPR for Pathological also benefitted from augmentation in the same dataset. Cha et al., instead of normalization, used the approach of merging similar diagnostic classes to reach balance between diagnostic groups, but the effect of this merging was not analysed in their study [14]. To our knowledge, the effect of balancing classes in otological studies, by any method, has never been investigated before, nor has the effect of augmentation of training data been studied, although the modifications are frequently performed [12,13,14]. Our finding of improvement in the performance for Pathological in dataset 1_norm_ up to a certain level of augmentation, but not thereafter, might be due to more Pathological images being used for training and therefore a better outcome of the augmentation. A subsequent decrease in accuracy when the augmentation factor is higher might be due to usage of too many similar images, or so-called overfitting of the system.

Sensitivity for abnormal (Pathological or Wax) was 100% for augmentation factor 5 in scenario A and B. This indicates that a Normal output from the CNN is most likely true, which is crucial for a screening system. Of clinical interest also is the finding that, in the separate sub-analysis, the CNN classified all nine NPD images as abnormal. This finding indicates that ears difficult to diagnose will most likely be classified as abnormal and as cases for further clinical assessment.

Overall accuracy for the non-normalized dataset was lowered for most augmentation factors when more images were added for training (scenario C and D), and normalization of the new dataset had no major influence. The added images in dataset 2 originate from various clinical materials and were captured with different otoscopes and endoscopes [11]. Furthermore, the reference standard for these images was established with a less solid method as compared to the expert panel’s diagnoses of dataset 1. These factors may have contributed to the lower accuracy for scenario C and D. The added images also included more variants of pathological ears. The higher prevalence of Pathological in dataset 2 (53%) compared to dataset 1 (16%) might also have contributed to the lowered accuracy of the CNN in scenario C and D, given the assumption that Normal is easier to diagnose for the CNN. The TPR for Pathological is, however, similar across augmentation factors between scenario A and C, even though scenario C contains more pathological samples. The imbalance in diagnostic groups should benefit scenario C’s TPR for Pathological, since CNNs are usually biased towards larger diagnostic groups. The use of only one type of digital otoscope in dataset 1 and the more solid reference standard are probable explanations for the similar performance.

GoogleNet was chosen because of its relative simplicity and overall good performance. In a study by Cömert, the overall accuracy for a selection of CNNs classifying five diagnoses was reported to be between 0.88 and 0.93 and GoogleNet showed an accuracy of 0.89 [29]. In our study the overall accuracy with GoogleNet was between 0.95 and 1.00 for three diagnoses.

In this study, we used three diagnostic groups to screen for pathological tympanic membranes. Cha et al. had six diagnostic classes and an accuracy of 0·94 [14] and Livingstone et al. reported an accuracy of 0·89 using 14 diagnostic classes [17]. Kuruvilla et al. used three classes and reached an overall accuracy 0·90 [7]. Using fewer diagnostic groups in our CNN might have contributed to the overall higher accuracy. The number of images used in our study is lower compared to the study by Cha et al. [14], Livingstone et al. [15] and Viscaino et al. [16] with 10,544, 724 and 720 images, respectively. Increasing the number of images used for training should theoretically result in better accuracy of a CNN. Our higher accuracy is partly explained by fewer diagnostic classes compared to the study by Cha et al. [14]. Both Cha et al. [14] and Livingstone et al. [12] utilised only one device for image capturing, as we did for dataset 1, so that does not explain the difference in accuracy.

It is difficult to compare our findings with previous studies, since the reference standard and number of diagnostic groups varies widely. Furthermore, the machine learning approaches differs between studies. The main, discernable strength of our study is our reference standard. Furthermore, from a clinical point of view, a CNN for screening purposes with high accuracy and sensitivity seems to have better value compared to a more intricate classification system for multiple diagnoses. It is also important to focus on clinically significant concerns. The need for assisted diagnostics is largest where the number of doctors is insufficient. It is in this context important to have a high sensitivity in identifying abnormal eardrums, which was the case in our study.

The number of tympanic membrane images was fairly low which could have influenced the accuracy. There is, however, a statistical uncertainty in our results due to the low image number. Adding images captured with various imaging techniques (dataset 2) decreased the overall accuracy, but sensitivity and specificity for abnormal ears were still very high. An optimal AI system must be able to handle different image sources. Furthermore, our reference image bank had an insufficient number of AOMs to be generalisable for a typical primary health care setting in a high-income country, where AOM is relatively common and CSOM less frequent. However, in an underserved, low-income setting this situation is commonly reversed, with more CSOM than AOM. AOM, however, is likely to be classified as Pathological and hence does not affect the accuracy. In the study, overall, we estimate that the risk of bias is low. Since images were representative of a low-income community, a more generalised use in other contexts would need additional images from such settings. Furthermore, actively draining ears were excluded in this study, but such ears are common at primary health care clinics, particularly in low-income countries. It is uncertain how the CNN would handle these.

Using only one CNN model in our study is a limitation, and further studies are needed to evaluate the effect of augmentation and normalization on more CNN models and with repeated tests.

In the future, a CNN screening system with high sensitivity and specificity for otitis media can be used by healthcare workers in rural or low-income areas where there is a lack of doctors. It could also alleviate doctors by having, for example, a nurse to carry out the first assessment of patients with ear complaints, guided by a CNN.

The screening CNN investigated in this study would benefit from minor updates before it is suitable for clinical usage. A larger training set, as already discussed, would be desirable. Automatic approval of the image quality prior to CNN analysis, with a possibility for the otoscopist to capture a new image if the first one is too poor, could enhance usability. A CNN that could include data of symptoms from a short questionnaire (e.g., fever or ear pain) may serve to further refine diagnostic classifications, when the symptoms are well underpinned for each specific diagnosis. It is important to emphasise that for all screening schemes, the logistics need to be in place (e.g., webservers and designated referral paths). Furthermore, a screening system must be studied in different clinical settings where diagnostic panoramas differ. In an era of rapid technological evolution, it is also worth mentioning the importance of validation of medical assistive tools before implementing them into the health care system, and to separate solutions with clinical evidence from those without.

## 5. Conclusions

The machine learning approach in this study presented high accuracy in screening for otitis media on images labelled by an expert panel. Sensitivity and specificity in discriminating normal from abnormal ears were satisfactory. Augmentation showed a slight positive effect on overall accuracy for normalized datasets, but we found no apparent advantage of normalization. A CNN like ours could be suitable to assist in basic screening for otitis media in underserved settings.

## Figures and Tables

**Figure 1 diagnostics-12-01318-f001:**
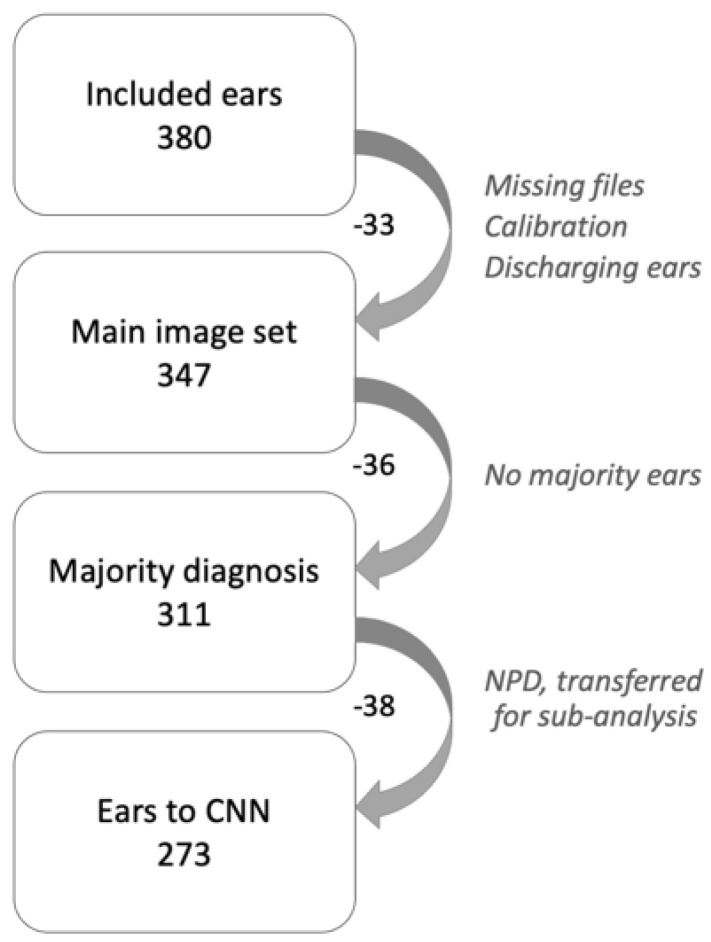
Flow chart. Description of tympanic membrane images from inclusion to selection for the convolutional neural network.

**Figure 2 diagnostics-12-01318-f002:**
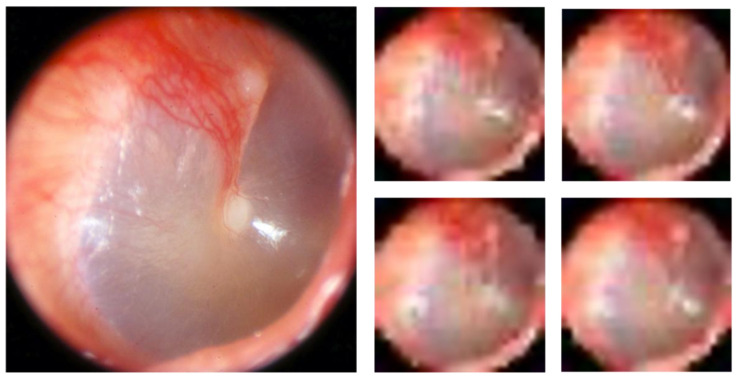
Augmentation. Augmentation demonstrated by generating multiple low-resolution images from a high-resolution image. In this example a number of 32 × 32 pixels images (to the right) were generated from one 1280 × 1024 pixels image (to the left) using an augmentation factor of 5.

**Figure 3 diagnostics-12-01318-f003:**
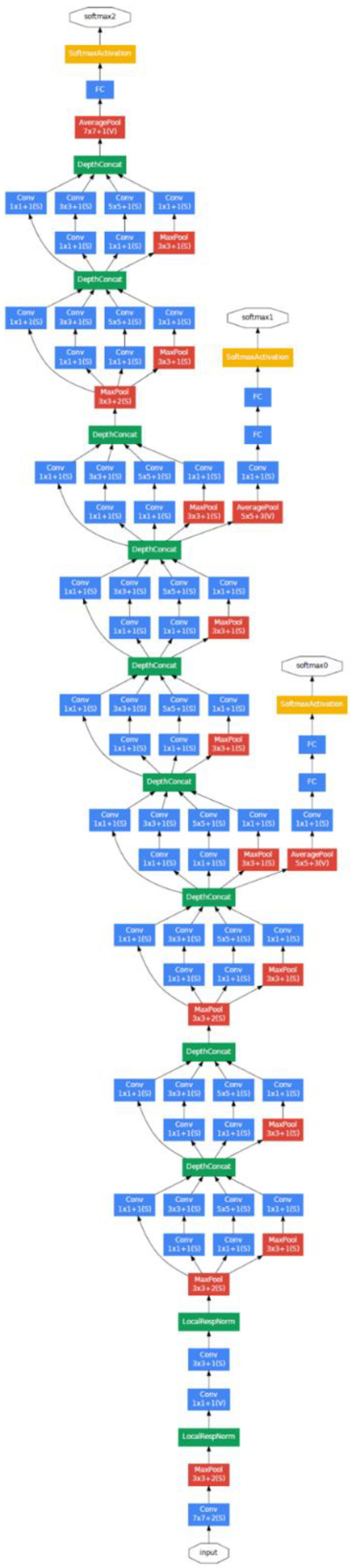
GoogLeNet architecture (from [25]).

**Figure 4 diagnostics-12-01318-f004:**
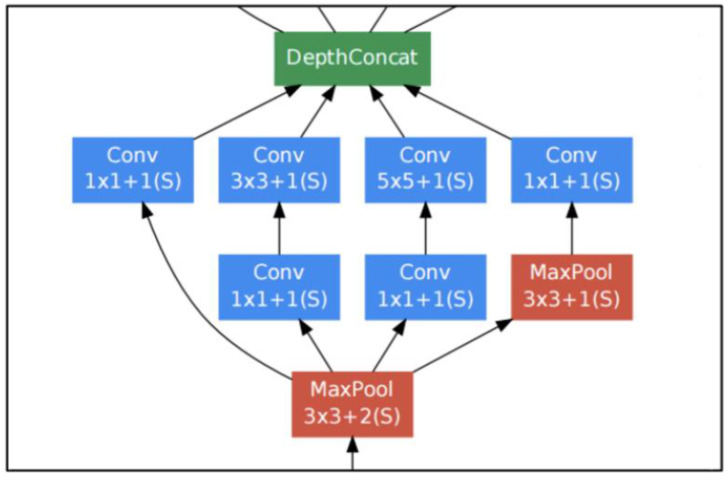
Inception modules (from [25]).

**Figure 5 diagnostics-12-01318-f005:**
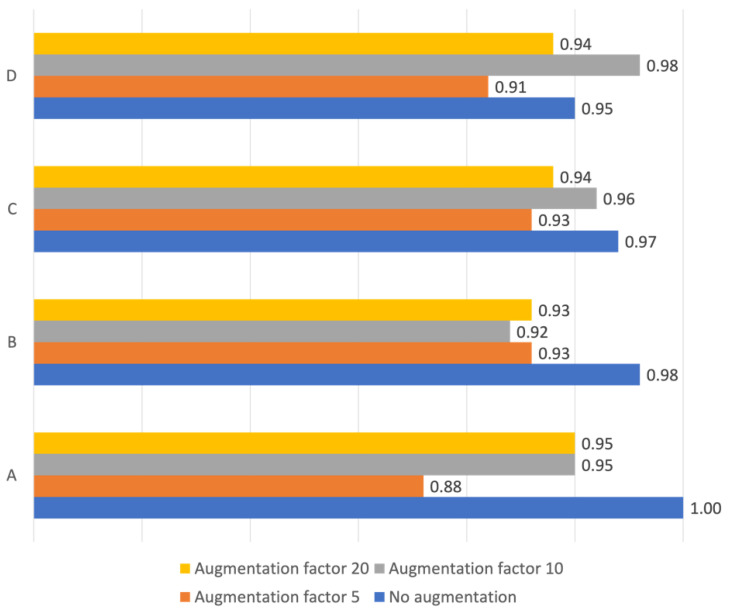
Overall accuracy. Overall accuracy for the convolutional neural network for scenarios (**A**–**D**) for augmentation factors 0, 5, 10 and 20. (**A**): training and testing on dataset 1, (**B**): training and testing on dataset 1_norm_, (**C**): training on dataset 1 and dataset 2 combined and testing on dataset 1, (**D**): training on dataset 1_norm_ and dataset 2_norm_ combined and testing on dataset 1_norm_.

**Figure 6 diagnostics-12-01318-f006:**
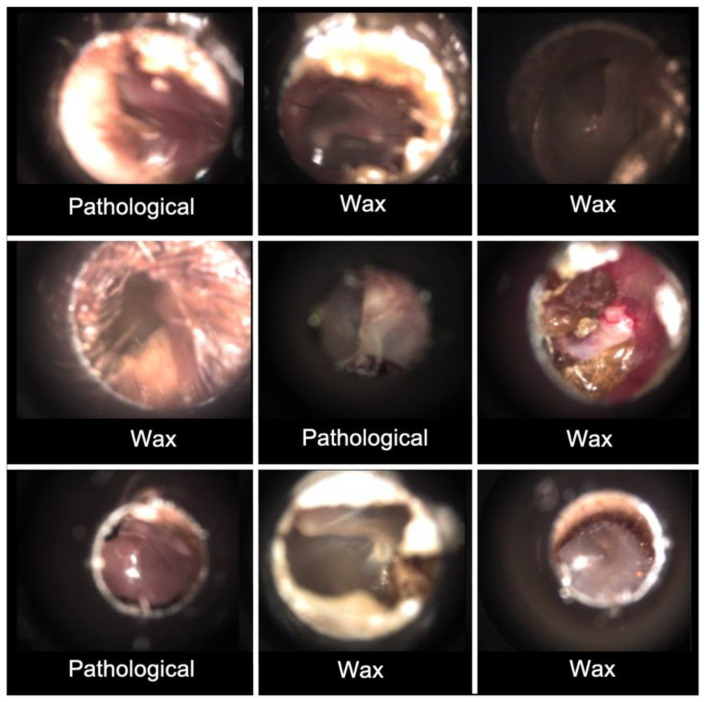
CNN diagnosis of NPD images. Convolutional neural network (CNN) diagnosis of nine not possible to determine (NPD) images when trained on dataset 1 without normalization and augmentation.

**Table 1 diagnostics-12-01318-t001:** Agreement (number and percentage) among expert panel members on diagnostic categories of images presented to the convolutional neural network (*n* = 273) and of the not possible to determine (NPD) images (*n* = 38).

	5 Out of 5	4 Out of 5	3 Out of 5	Total (*n*)
**AOM**	0 (0%)	0 (0%)	2 (100%)	2
**CSOM**	15 (75%)	5 (25%)	0 (0%)	20
**Normal**	90 (49%)	48 (26%)	45 (25%)	183
**OME**	1 (4%)	4 (18%)	17 (77%)	22
**Wax**	27 (59%)	9 (19%)	10 (22%)	46
**Total***	133 (49%)	66 (24%)	74 (27%)	273
**NPD**	9 (24%)	10 (26%)	74 (27%)	38

The number of expert panel members that agreed upon the diagnosis is referred to as 5 out of 5, and so forth. Total* refers to the 273 images analysed by the convolutional neural network.

**Table 2 diagnostics-12-01318-t002:** Confusion matrix for scenario A and B. 1= perfect correspondence and 0 = no correspondence.

	Norm (0)	Norm (5)	Norm (10)	Norm (20)	Path (0)	Path (5)	Path (10)	Path (20)	Wax (0)	Wax (5)	Wax (10)	Wax (20)
Normal	**1.0** **1.0**	**1.0** **1.0**	**1.0** **1.0**	**1.0** **1.0**	0 0	0 0	0 0	0 0	0 0	0 0	0 0	0 0
Path	0 0.07	0 0	0.10 0	0.15 0	**1.0** **0.72**	**0.90** **0.79**	**0.81** **0.96**	**0.78** **0.86**	0 0.21	0.10 0.27	0.09 0.04	0.07 0.14
Wax	0 0	0 0	0 0	0 0	0 0	0.05 0	0.01 0.15	0 0.18	**1.0** **1.0**	**0.95** **1.0**	**0.99** **0.84**	**1.0** **0.82**

Combined confusion matrix presenting the true positive rate (grey fields) and false positive rate (white fields), for scenario A and B for augmentation factors 0, 5, 10 and 20 (augmentation factor within brackets). In each box, the value above is for scenario A and the value below is for scenario B. Abbreviations: Normal (Norm), Pathological (Path).

**Table 3 diagnostics-12-01318-t003:** Sensitivity and specificity.

SCENARIOS		A	B	C	D
**AUGM 0**	Sensitivity	100%	98%	100%	98%
	Specificity	100%	100%	100%	100%
**AUGM 5**	Sensitivity	100%	100%	100%	100%
	Specificity	100%	100%	100%	100%
**AUGM 10**	Sensitivity	95%	100%	96%	100%
	Specificity	100%	100%	100%	100%
**AUGM 20**	Sensitivity	93%	100%	100%	100%
	Specificity	100%	100%	100%	100%

Sensitivity and specificity for the convolutional neural network in identifying ears with pathology or wax, all scenarios and with different augmentation (Augm) factors.

**Table 4 diagnostics-12-01318-t004:** Confusion matrix for scenario C and D. 1 = perfect correspondence and 0 = no correspondence.

	Norm (0)	Norm (5)	Norm (10)	Norm (20)	Path (0)	Path (5)	Path (10)	Path (20)	Wax (0)	Wax (5)	Wax (10)	Wax (20)
Norm	**1.0** **1.0**	**1.0** **1.0**	**1.0** **1.0**	**1.0** **1.0**	0 0	0 0	0 0	0 0	0 0	0 0	0 0	0 0
Path	0 0.11	0 0	0.07 0	0 0	**0.59** **0.90**	**0.83** **0.83**	**0.86** **0.87**	**0.87** **0.92**	0.41 0	0.17 0.17	0.08 0.13	0.13 0.08
Wax	0 0	0 0	0 0	0 0	0 0.06	0.30 0.13	0.02 0.07	0 0.12	**1.0** **0.94**	**0.70** **0.87**	**0.98** **0.93**	**1.0** **0.88**

Combined confusion matrix presenting the true positive rate (grey fields) and false positive rate (white fields), for scenarios C and D for augmentation factors 0, 5, 10 and 20 (augmentation factor within brackets). In each box, the value above is for scenario C and the value below is for scenario D. Abbreviations: Normal (Norm), Pathological (Path).

## Data Availability

All data are fully available without restriction. The data underlying the results presented in the study are available from: https://doi.org/10.6084/m9.figshare.14376842 and https://doi.org/10.6084/m9.figshare.14420615 (accessed on 12 April 2021).

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
