# Peer review of "A Machine Learning Approach to Screen for Otitis Media Using Digital Otoscope Images Labelled by an Expert Panel"

_diagnostics, 2022, doi:10.3390/diagnostics12061318_

Round 1

Reviewer 1 Report

This paper presents a machine learning-based approach to detect otitis media using digital otoscope images labeled by an expert panel. In machine learning-based methods, an important element is the learning data, making the effectiveness of the prediction and the thesis largely dependent on the selection of criteria and the amount of data and the appropriate selection of data. 

Minor remarks:
1) I suggest describing in more detail on what basis and from what the scope of input data selection in terms of algorithm learning efficiency. 
2) The authors could refer to other machine learning methods and compare the presented results with other achievements.
3) The article needs minor language corrections.

Author Response

1) I suggest describing in more detail on what basis and from what the scope of input data selection in terms of algorithm learning efficiency. - Changes have been made in the manuscript in response to the constructive comment, see row 225-233.”Instead of normalizing the datasets (or undersampling by removing images) and applying data augmentation (a form of oversampling by creating new images) as described in Section 2.4, an alternative approach could have been taken to ensure that no data are discarded. Instead of data resampling, the classifier could have been modified by applying thresholding, learning rate adjustment or adjusting prior class probabilities.”

2) The authors could refer to other machine learning methods and compare the presented results with other achievements. -  The focus of our study was to compare what effect normalization and augmentation would have on classifying tympanic membrane images diagnosed by 5 ENTs. Since our study the models have been refined further. We have added a section in Discussion, row 415-419 GoogleNet was chosen because of its relative simplicity and overall good performance. In a study by Zafer et al., the overall accuracy for a selection of CNNs classifying five diagnoses was reported  to be between 0.88 and 0.93 and GoogleNet showed an accuracy of 0.89. In our study the overall accuracy with GoogleNet was between 0.95 and 1.00 for three diagnoses.”

3) The article needs minor language corrections. - The manuscript has been reviewed and updated to correct any language and minor spelling errors.

Reviewer 2 Report

This paper studies digital otoscope image classification using a CNN approach. This is a good practical study; however, some technical details need further clarification.

1. The CNN image classification model is trained from a well-balanced dataset, as the re-sampling is applied as a pre-processing to balance the data. It states that "the number of images in each diagnostic group in the normalized data sets is equal to the group with the lowest number of images." Such a model trained from a well-balanced dataset might not be useful as it is inbalanced in practice. Therefore, it would be good to study how to address the inbalance issue via revising the model (e.g., changing the loss), instead of applying data re-sampling.

2. Only one model, Inception v1, is tested in the paper. It would be good to evaluate more SOTA CNN models, such as ResNet, DPN, etc.

3. Significant details of model building are missing, such as the optimizer, batch size, etc.

Author Response

  1. The CNN image classification model is trained from a well-balanced dataset, as the re-sampling is applied as a pre-processing to balance the data. It states that "the number of images in each diagnostic group in the normalized data sets is equal to the group with the lowest number of images." Such a model trained from a well-balanced dataset might not be useful as it is inbalanced in practice. Therefore, it would be good to study how to address the inbalance issue via revising the model (e.g., changing the loss), instead of applying data re-sampling. - The scope of our study was to study how well a CNN could screen for otitis media and ear pathology in comparison to a reliable reference standard. We do not present results from an imbalanced dataset with the learning rate adjusted for each pathology.
  2. Only one model, Inception v1, is tested in the paper. It would be good to evaluate more SOTA CNN models, such as ResNet, DPN, etc. - As indicated above in comment 2 to Reviewer 1 the focus of our study was to compare what effect normalization and augmentation would have on classifying tympanic membrane images diagnosed by 5 ENTs. There were only a few freely available CNNs to choose from at the time of the study, but since then the models have developed and been further refined. New models must be evaluated in other studies.
  3. Significant details of model building are missing, such as the optimizer, batch size, etc. - Changes have been made accordingly in the manuscript, row 197-205 together with the addition of two new figures (figure 3 and 4 in the revised document). Figure 3 shows the structure of this CNN. It consists of structure combinations of convolutional layers (large blue rectangles), dense layers (green) , pooling layers (red), and activation/decision layers (yellow). Figure 4 shows the inception module architecture with its various convolutional and pooling blocks. The standard Tensorflow implementation was used without any architecture modification. The batch size was 128. It used asynchronous stochastic gradient descent for training with an initial learning rate of 0.1 and an exponential decay factor of 0.1. The learning rate was updated every 50 epochs. Image data were reformatted to conform to the required input data format.”

Round 2

Reviewer 2 Report

Thanks for the revision. Regarding the point #2, the authors stated that "the focus of our study was to compare what effect normalization and augmentation would have on classifying tympanic membrane images diagnosed by 5 ENTs. There were only a few freely available CNNs to choose from at the time of the study, but since then the models have developed and been further refined. New models must be evaluated in other studies."
-> I think it is still needed to do more experiments to test what effect normalization and augmentation would have on MULTIPLE CNN models. Otherwise, it might be only effective for the Inception v1 model.

Author Response

Thank you for considering the revised manuscript and for the insightful comment. We agree that a study on multiple CNN models would be the best way to study the effect of augmentation and normalization together with repeated measurements. Our main aim for this study however was to investigate a CNNs accuracy in screening for otitis media based on images classified in a robust and structured manner with a panel of experts to ensure high quality ground truth data. The primary study aim is stated as "to investigate the performance of a machine learning approach (a CNN) for otitis media screening using digital otoscopic images of tympanic membranes diagnosed by an expert panel."  A secondary aim was to also consider the effect of augmentation and normalization on the accuracy in our model. Whilst we agree that a study of augmentation and normalization across other CNNs would be very interesting and valuable this study did not propose this as its main objective. The strength of our study lies in its clinically oriented focus together with a robust reference standard that could be used as a method for future AI otitis media studies. To address the matter of future studies to explore augmentation and normalization we have now added this as a limitation and recommend future investigations to explore this.

To highlight this in the manuscript we added the following in Discussion and the section of limitations, row 457-459:

Using only one CNN model in our study is a limitation, further studies are needed to evaluate the effect of augmentation and normalization on more CNN models and with repeated tests.

Kind regards

Round 3

Reviewer 2 Report

thanks for your reply.